# The Potential of Plant-Based Bioactive Compounds on Inhibition of Aflatoxin B1 Biosynthesis and Down-regulation of *aflR*, *aflM* and *aflP* Genes

**DOI:** 10.3390/antibiotics9110728

**Published:** 2020-10-23

**Authors:** Nassim Safari, Mehran Mirabzadeh Ardakani, Roghayeh Hemmati, Alessia Parroni, Marzia Beccaccioli, Massimo Reverberi

**Affiliations:** 1Department of Plant Protection, Faculty of Agriculture, University of Zanjan, Zanjan 45371-38791, Iran; rhemati@znu.ac.ir; 2Department of Environmental Biology, Sapienza University, 00185 Roma, Italy; alessia.parroni1@gmail.com (A.P.); marzia.beccaccioli@uniroma1.it (M.B.); 3Department of Traditional Pharmacy, Faculty of Pharmacy, Tehran University of Medical Science, Tehran 14176-14411, Iran; mehranmirab@yahoo.com

**Keywords:** postharvest disease control, plant extracts, *Aspergillus flavus*, aflatoxin B1, aflatoxin biosynthesis, gene regulation

## Abstract

The use of plant extracts in pre- and post-harvest disease management of agricultural crops to cope with aflatoxin B1 contamination has shown great promise due to their capability in managing toxins and safe-keeping the quality. We investigated the anti-aflatoxigenic effect of multiple doses of eight plant extracts (*Heracleum persicum*, *Peganum harmala*, *Crocus sativus*, *Trachyspermum ammi*, *Rosmarinus officinalis*, *Anethum graveolens*, *Berberis vulgaris*, *Berberis thunbergii*) on *Aspergillus flavus* via LC-MS and the down-regulatory effect of them on *aflR*, *aflM* and *aflP* genes involved in the aflatoxin B1 biosynthesis pathway using RT-qPCR analyses. Our results showed that *H. persicum* (4 mg/mL), *P. harmala* (6 mg/mL) and *T. ammi* (2 mg/mL) completely stopped the production of aflatoxin B1, without inducing significant changes in *A. flavus* growth. Furthermore, our findings showed a highly significant correlation between the gene expression and the aflatoxin B1 biosynthesis, such that certain doses of the extracts reduced or blocked the expression of the *aflR*, *aflM* and *aflP* and consequently reduced the synthesis of aflatoxin B1. Interestingly, compared to the regulatory gene (*aflR*), the down-regulation of expression in the structural genes (*aflM* and *aflP*) was more consistent and correlated with the inhibition of aflatoxin B1 production. Overall, this study reveals the anti-aflatoxigenic mechanisms of the selected plant extracts at the gene expression level and provides evidence for their use in plant and crop protection.

## 1. Introduction

A wide variety of food crops are contaminated with mycotoxins, which occur due to pathogenic fungi in the pre- and post-harvest stages [1]. Among the mycotoxins, aflatoxins represent a serious threat to human and livestock health as well as to international trade. Aflatoxins are potent toxic, carcinogenic, mutagenic, immunosuppressive agents that are mainly produced as secondary metabolites by fungal species belonging to the *Aspergillus* section *Flavi* on a wide range of food products [2]. Among several different types of aflatoxins, aflatoxin B1 is the most common carcinogenic and can be found in foods, such as cereals, corn, rice, peanuts, and pistachios [3]. In addition to serious health risks, toxigenic fungi impose significant annual economic losses to food products [4]. Given that global climate change and global warming are boosting the spread of mycotoxin-producing fungi, mycotoxin contamination is expected to increase in the future [5,6].

Nowadays, chemical and physical methods are commonly used to control or eliminate aflatoxins. These methods, however, may lead to accumulation of harmful chemical compounds on food products and even reduce their nutritional values [1,7]. Thus, exploration of alternative safer strategies has gained great interest. Several studies have focused on biological approaches such as using non-toxic strains of *Aspergillus*, competitive yeasts, and bacteria or fungi, and also the use of herbal extracts to cope with toxin contamination [8,9]. Although prevention of aflatoxin contamination by inhibiting fungal growth in stored grains, food and feed is often considered an optimal approach, other techniques such as inhibiting the production of aflatoxins or favoring their decomposition are also necessary, as the aflatoxin contamination is often inevitable. The use of plant-based active ingredients (essential oils, plant extracts, and isolated molecules) instead of chemical compounds is usually preferred because they are often safer, environmentally friendly and effective in controlling the growth of fungi as well as eliminating toxic compounds with different mechanisms [10,11,12,13]. Their antibacterial activity, antioxidant potential, and strong organoleptic properties gives them broad applications as medicines, food and feed additives, cosmetics and perfume ingredients [14]. Furthermore, these compounds can be effective in pre- and post-harvest control [15].

The aflatoxin B1 biosynthesis pathway, one of the known pathways of secondary fungal metabolism, is affected by various environmental factors [16]. Aflatoxin biosynthesis is supported by 30 structural genes and at least 27 enzymatic reactions, in which *aflR* and *aflS* act as regulatory genes [17,18]. External regulator genes (*LaeA*, *VeA*) have also been shown to increase expression of the genes of the aflatoxin biosynthesis cluster [19,20,21]. Several recent studies have investigated the gene expression mechanism of plant-based compounds and have reported their effectiveness in suppressing the expression of the aflatoxin biosynthesis genes and thereby controlling aflatoxin biosynthesis [22,23,24,25]. For example, Liang et al. (2015) reported that cinnamaldehyde (0.40 mM/L), eugenol (0.80 mM/L), and citral (0.56 mM/L) reduced aflatoxin B1 production and suppressed the expression of *aflR*, *aflT*, *aflD*, *aflM*, and *aflP* genes in *A. flavus* [22]. Caceres et al. (2016) investigated the inhibitory effect of eugenol using a large-scale q-PCR approach in the production of AB1 and showed that eugenol at a concentration of 0.5 mM controlled the expression of 26 out of 27 genes in the aflatoxin biosynthesis gene cluster [24]. Another study carried out by El Khoury et al. (2017) showed that 10 mg/mL of hyssop aqueous extract repressed expression of all genes belonging to aflatoxin B1 cluster genes and also fifteen regulatory genes including *veA* and *mtfA* [25]. Caceres et al. (2017) showed that piperine significantly inhibited 25 out of 27 genes involved in the aflatoxin biosynthetic pathway and reduced a global regulator gene (*veA*) [23].

In recent years, there has been growing interest in using natural compounds in the human diet due to increased consumer awareness about the negative effects of pesticides and artificial additives [26]. However, despite the various studies conducted, many plant species are still unexplored and unknown, and the search for introducing more potent antimicrobial plant agents is ongoing [27]. Thus, further research is needed to explore safe and natural compounds that have the potential to control aflatoxins and then demonstrate the underlying mechanism. The objective of this study was to investigate the inhibitory effect of medicinal herbs selected from traditional Iranian medicine sources. To this end, we selected eight natural plant extracts that are less well or not studied or little is known about the molecular mechanism of their aflatoxin inhibition and the affected aflatoxin pathways, but could be potential sources of controlling aflatoxin production in agricultural products. We considered their pleasantness as a food additive and their affordability, and examined their anti-aflatoxigenic effects at the molecular level by studying some of the key genes involved in the aflatoxin B1 biosynthesis pathway. Two structural genes (*aflM* and *aflP*) and one regulatory gene (*aflR*) that are involved in aflatoxin biosynthesis, were examined.

As we describe below, we found that the effective doses of the selected plant extracts can control or stop aflatoxin B1 production and concluded that the decrease in aflatoxin B1 biosynthesis can be due to the suppressed expression level of the studied genes.

## 2. Results

### 2.1. Effective Concentration of Plant Extracts Can Control the Growth of A. flavus and Inhibit the Production of Aflatoxin B1

To select a suitable culture medium for *A. flavus* growth and aflatoxin B1 production, we investigated six growth mediums, including: Potato Dextrose Broth (PDB), Sabouraud Dextrose Broth (SDB), Czapek Yeast Extract (CYE), Malt Extract Broth (MEB), Yeast Extract Peptone Dextrose (YEPD), and Yeast Extract Sucrose (YES). For each medium, we obtained the fungal growth using the dry weight of the mycelium and the aflatoxin B1 production using the HPLC-MS/MS analysis.

First, we assessed whether or not the choice of culture medium influences the results by performing a multivariate analysis of variance (MANOVA), where we specified aflatoxin B1 production and fungal growth as dependent variables and the culture medium as an independent variable. The culture medium was defined as a categorical variable. The MANOVA revealed a significant impact of culture medium on aflatoxin B1 production and the fungal growth (*f*-value = 6.9, *p* < 0.001). Next, to find the medium culture that offers an optimal environment for the fungal growth and aflatoxin B1 production, we carried out analysis of variance (ANOVA) on both fungal growth and aflatoxin B1 production, each separately, as a function of culture medium. The culture medium was specified as a categorical variable. ANOVA showed that fungal growth and aflatoxin B1 production are significantly different across culture media (fungal growth: *f*-value = 13.3, *p* << 0.001; aflatoxin B1 production: *f*-value = 38, *p* << 0.001). Figure 1 displays the mean (bars) and standard deviation (error bars) of fungal growth and aflatoxin B1 production for each culture medium. ANOVA tests if the results are significant overall but it provides no information about where those differences exactly lie. To find these differences, we used a post hoc Tukey’s Honest Significant Difference (HSD) test to find out which specific groups’ (culture media) means (compared with each other) are different. To test all pairwise comparisons among means using the Tukey HSD, we calculated HSD for each pair of means, separately for fungal growth and aflatoxin B1 production. Appendix A displays the results of pairwise comparisons of fungal growth and aflatoxin B1 production between different culture media. Our analysis showed that, compared to other culture media, the Yeast Extract Sucrose (YES) medium provides a highly significant condition for fungal growth and aflatoxin B1 production.

Subsequent to picking the YES medium as an optimal growth medium for our experiments, we sought to explore the inhibition of the aflatoxin B1 biosynthesis pathway by means of medicinal herbs considering their antimicrobial properties and their pleasantness as a food/feed additive. Eight medicinal herbs including *H. persicum*, *P. harmala*, *C. sativus*, *T. ammi*, *R. officinalis*, *A. graveolens*, *B. vulgaris*, and *B. thunbergii* were selected. We tested different doses of plant extracts and used those doses that showed no significant influence on fungal growth (*p* > 0.05). The selected doses were: 1, 2, and 4 mg/mL of *H. persicum*; 2, 4 and 6 mg/mL of *P. harmala*; 4, 6 and 8 mg/mL of *C. sativus*; 0.5, 1 and 2 mg/mL of *T. ammi*; 1, 2, and 4 mg/mL of *R. officinalis*; 2, 4 and 6 mg/mL of *A. graveolens*, and 4, 6 and 8 mg/mL of *B. vulgaris* and *B. thunbergii*. The amount of fungal growth and aflatoxin B1 production were obtained for all tests. To statistically assess the impact of these extracts on aflatoxin B1 production, we compared the aflatoxin B1 production between the control test and all treatment conditions (different plant extracts and doses) using a two-sample *t*-test (Figure 2). Our analysis showed that *H. persicum* (2 and 4 mg/mL; *t*-value = 16.4, and 17.5), *P. harmala* (4 and 6 mg/mL; *t*-value = 14.1, 17.5), *T. ammi* (1 and 2 mg/mL, *t*-value = 17.2, 17.5), *R. officinalis* (4 mg/mL, *t*-value = 16.6), *A. graveolens* (4 and 6 mg/mL; *t*-value = 14.8, 16.9), and *B. thunbergii* (8 mg/mL, *t*-value = 15.9) significantly reduced the production of aflatoxin B1 (*p* < 0.001). Moreover, *H. persicum* (1 mg/mL, *t*-value = 14.2), *C. sativus* (8 mg/mL, *t*-value = 13.8), *T. ammi* (0.5 mg/mL, *t*-value = 11.5), *R. officinalis* (1, 2 mg/mL, *t*-value = 11.7, 11.5), *A. graveolens* (2 mg/mL; *t*-value = 11.4), and *B. vulgaris* (8 mg/mL, *t*-value = 13.7) significantly inhibited the aflatoxin B1 biosynthesis at the significance level of less than 0.01. Interestingly, the aflatoxin production was fully eliminated in *H. persicum* (4 mg/mL), *P. harmala* (6 mg/mL) and *T. ammi* (2 mg/mL) tests. Furthermore, the inhibition of aflatoxin B1 production in *P. harmala* (2 mg/mL), *C. sativus* (6 mg/mL), *B. vulgaris* (6 mg/mL) and *B. thunbergii* (4 and 6 mg/mL) was significant (*p* < 0.05).

Next, we applied a post-hoc analysis to examine the inhibition of aflatoxin production resulting from addition of different doses of plant extracts while statistically controlling for the potential effect of fungal growth. For this purpose, we used a general linear model (GLM), where we specified the aflatoxin B1 production as a dependent variable, and a categorical variable (comprising two levels: “control”, and “treatment”) plus fungal growth as independent variables (aflatoxin B1 production ~ groups + fungal growth). This analysis was applied between each treatment condition and the control. In essence, this GLM analysis applies a two-sample *t*-test between each treatment condition and the control, but controls for the potentially confounding impact of fungal growth. Figure 3 displays the results for all treatment conditions. Each bar represents the *t*-value obtained from GLM analysis for all treatment conditions. The result demonstrates a significant negative effect of plant extracts on aflatoxin B1 production irrespective of fungal growth.

### 2.2. Addition of Extract Concentrations Leads to Gene Expression Reduction and Thereby Aflatoxin B1 Control

To answer the question whether or not the inhibition of aflatoxin B1 resulting from addition of extract concentrations occurs at the transcriptomic level, we investigated the expression of a regulatory gene (*aflR*) and two structural genes (*aflM* and *aflP*) that are known to be involved in the aflatoxin biosynthetic pathway [18]. These tests were conducted for various doses of several plant extracts and the fold-change reductions in expression levels compared to control conditions for the noted genes were obtained. Our analysis showed a highly significant fold-change reduction in expression level of *aflP* following the addition of *H. persicum*, 4 mg/mL (265.2 times, *p* < 0.001), *P. harmala*, 6 mg/mL (98.4 times, *p* < 0.001), *C. sativus*, 8 mg/mL (49.6 times, *p* < 0.001), *T. ammi*, 2 mg/mL (378.7 times, *p* < 0.001), *R. officinalis*, 4 mg/mL (62.2 times, *p* < 0.001), *A. graveolens*, 6 mg/mL (282.7 times, *p* < 0.001), *B. vulgaris*, 8 mg/mL (16.5 times, *p* < 0.001), and *B. thunbergii*, 8 mg/mL (31.9 times, *p* < 0.001) at the maximum effective dose of extracts, and thus reached a complete inhibition of aflatoxin B1. Analogous treatments performed on the selected 8 extracts identified significant down-regulation of *aflM* (respectively: 57.5, 17.6, 19.8, 109.6, 18.9, 30.4, 3.7, 11.2 times (all *p* < 0.001)) and marginally significant down-regulation of *aflR* (respectively: 9.1, 2.6, 1.9, 6.6, 2.0, 8.6, 1.4, 2.6 times (all 0.01 < *p* < 0.001)). The results are depicted in Figure 4, where each bar is colored based on gene type and shows the fold change expression for different doses of extracts. Overall, our finding that addition of the plant extracts leads to reduction of the gene expression, and thereby inhibition of aflatoxin B1 production, could suggest a gene expression mechanism underpinning the aflatoxin B1 control.

Subsequently, we performed a correlation analysis between the aflatoxin B1 production and fold change expression of three genes, separately, where all plant extracts and doses were included. Our result showed a significant correlation between the percentage of aflatoxin B1 production and the fold change of gene expression; however, the correlation was dependent on the gene type, where down-regulation of expression was more significant for the structural genes (*aflM: r* = 0.84, *p* << 0.001; and *aflP: r* = 0.87, *p* << 0.001) compared to the regulatory gene (*aflR: r* = 76, *p* < 0.001) though both were highly significant (Figure 5). Interestingly, for up to 0.25-fold of *aflR* expression, almost no aflatoxin B1 was produced and this inconsistent relationship was also observed for other expression values (e.g., 0.37-fold and 0.77-fold). In contrast, for structural genes, particularly *aflP*, this relationship was closer and more consistent across all extracts and doses. This strong correlation between the expression of the structural genes (*aflM* and *aflP*) and the amount of aflatoxin B1 biosynthesis suggests an inhibition mechanism of the introduced plant extracts.

## 3. Discussion

In this study, we investigated the inhibition of aflatoxin B1 production when different concentrations of eight selected plant extracts (*C. sativus*, *T. ammi*, *H. persicum*, *A. graveolens*, *P. harmala*, *R. officinalis*, *B. vulgaris*, *B. thunbergii*) were added, using LC-MS and RT-qPCR analyses, to examine the mechanism of the antifungal and anti-toxigenic properties of the plant extracts against *A. flavus* and aflatoxin B1 production.

Results of the present study showed that *A. flavus* can grow in YES synthetic medium and produce aflatoxin B1 (Figure 1 and Appendix A) but the addition of certain doses of the selected extracts suppressed the production of mycotoxin, without inducing significant changes in *A. flavus* growth (Figure 2). Moreover, by increasing the dose of extracts, the expression of the studied genes was reduced or switched off, consequently the production of aflatoxin B1 was reduced. This finding indicates an association between results of LC-MS and RT-qPCR analyses, and thus represents a relationship among extract dose, gene expression, and aflatoxin B1 production.

The inhibition of aflatoxin B1 was dose-dependent and as shown in Figure 2, by increasing the concentration of extracts, despite the lack of significant reduction in fungal growth, a decrease in aflatoxin production was observed for all tests. The results of LC-MS analyses showed that the aflatoxin B1 production was fully eliminated in *H. persicum* (4 mg/mL), *P. harmala* (6 mg/mL) and *T. ammi* (2 mg/mL).

Aflatoxin B1 production is variable as a function of extract type and concentration, but it is independent of fungal growth as demonstrated by the GLM analysis (Figure 3). In this line, several reports have shown that not all the extracts have fungicidal activity [28].

Our analysis showed that when *A. flavus* was treated with different concentrations of plant extracts, the structural genes were more inhibited than the *aflR*. Moreover, for these structural genes, particularly *aflP*, the relationship between the aflatoxin B1 production and the gene expression was highly significant and very consistent across all plant extracts and concentrations. This strong correlation between the expression of the structural genes (*aflM* and *aflP*) and the amount of aflatoxin B1 biosynthesis may suggest a prominent and direct role of the selected structural genes in the mechanism of aflatoxin B1 inhibition by extracts.

Aflatoxin cluster genes are coordinately regulated and are in the genome as a single copy and their expression is coordinated by two cluster-specific regulators (*aflR* and *aflS*) [29]. Studies have shown the role of at least 27 enzymatic reactions in the process of aflatoxin synthesis. The biosynthesis of aflatoxin B1 in *A. flavus* is a complex and multi-step process that involves the enzymes of metabolic pathways and intermediates. Norsoloric acid, versicolorin A, sterigmatocystin and finally, aflatoxin B1 are more stable metabolites that are produced in this pathway [18]. In this study, we focused on a regulatory gene (*aflR*) and two pathway genes (*aflM* and *aflP*). The *aflR* gene, a positive regulatory gene for aflatoxin biosynthesis, encodes *AFLR* transcription factor (a protein containing a zinc-finger DNA-binding motif) [30]. The *aflM* (*ver-1*) gene is one of the four genes involved in the conversion of versicolorin A (VERA) into sterigmatocystin and is about the middle gene of the aflatoxin biosynthesis pathway gene cluster. The *aflM* has been predicted to encode a ketoreductase and is involved in the conversion of VERA to demethyl sterigmatocystin (DMST) [31,32,33]. The *aflP* (*omtA*) gene is located at the end of the aflatoxin biosynthesis pathway gene cluster and is involved in the conversion of sterigmatocystin (ST) into O-methyl sterigmatocystin (OMST), which is the precursor of aflatoxin B1, by coding O-methyltransferase [18,34].

Our results showed that addition of the selected plant extracts, especially all concentrations of *H. persicum* and *T. ammi*, led to a significant decrease of *aflM* expression (57.5 and 109.6 times, respectively), though the expression of *aflM* was completely inhibited upon addition of *T. ammi*, 2 mg/mL. This result could be related to the notion that these extracts, in their effective concentrations, can interfere with the enzyme that synthesizes the intermediate of sterigmatocystin in the middle of the aflatoxin biosynthetic pathway and hence, eventually, can induce inhibitory effects on the production of aflatoxin B1. The largest effect of the extracts was observed on *aflP*, such that the expression of this gene was significantly reduced in 18 out of 24 tests at *p* < 0.001. Indeed, expression of *aflP* was completely inhibited in *H. persicum* (2 and 4 mg/mL), *P. harmala* (6 mg/mL), *T. ammi* (2 mg/mL) and *A. graveolens* (6 mg/mL). This output is supported by previous studies reporting that expression of *aflP* was correlated with *A. flavus* ability to produce aflatoxins [18,35], and may indicate that the effective concentrates of extracts can lead to aflatoxin B1 inhibition by affecting the *aflP* and disrupting the production of one of the aflatoxin precursors at the end of the biosynthetic pathway. Overall, following the addition of plant extracts, the mRNA level of the structural genes (*aflM* and *aflP*) was significantly (*p* < 0.001) downregulated for most of the extract concentrations (Figure 4, Appendix A). In contrast, the *aflR* gene was less downregulated, although it was marginally significant for most concentrations (*p* ~ 0.01). The *aflR* gene encodes a Zn2Cys6-type transcription factor (*aflR*) that is necessary for transcription of most if not all of the genes in the aflatoxin cluster [36,37]. It is the main regulatory factor in the aflatoxin biosynthesis gene cluster and can be auto regulated [18,38]. Lack of *aflR* gene or abnormal *aflR* gene can be a strong indicator of a strain’s disability in aflatoxin production [30]. Casquete et al. evaluated three strains of aflatoxin-producing *Aspergillus* and showed that the relative expression of *aflR* and *aflS* genes is significantly correlated with the concentration of the produced aflatoxins [39]. Likewise, another study found that an increase in the expression of *aflR* could increase the production of aflatoxin B1 in *A. flavus* by up to 50 times [40]. Our finding that represents less downregulation of *aflR* compared with *aflP* and *aflM* could be explained by the point that the *aflR* is a regulatory gene, and is more stable compared with the other two genes; thus, even minor changes in its expression can affect other genes in the cluster and therefore can lead to a substantial decrease in the gene expression level of other genes. Furthermore, one may argue that the downregulation of these genes could depend on their chronological intervention in the biosynthetic pathway.

Although the effect of addition of extracts on gene expression and aflatoxin production is understood, it is not established how the relationship between gene expression and amount of aflatoxin varies across different genes. Indeed, our analysis determined a more consistent and stronger relationship between the aflatoxin B1 production and expression of structural genes (*aflM* and *aflP)* than that of *aflR* (see Figure 4 for more details) for the selected extracts and concentrations. As mentioned above, the regulatory feature of *aflR* may explain the observed low correlation and low consistency for *aflR* compared to other genes.

Our analysis showed a highly significant fold-change reduction in expression level (compared to control conditions) following the addition of *H. persicum* (*aflP*: 265.2 times, *aflM*: 57.5 times, *aflR*: 9.1 times), *P. harmala* (*aflP*: 98.4 times, *aflM*: 17.6 times, *aflR*: 2.6 times), and *T. ammi* (*aflP*: 378.7 times, *aflM*: 109.6 times, *aflR*: 6.6 times). According to this result, inhibition by herbal extracts occurs at the transcriptomic level as expression ratios of the studied genes in the aflatoxin cluster were severely reduced. Nonetheless, it should be noted that this does not mean that these genes are the direct molecular target of plant extracts; they may disrupt the aflatoxin production process through several cellular mechanisms, including cellular signaling, global transcription factors, and oxidative stress. The transcriptomic effect of these extracts and the correlation between the gene expression and production of secondary metabolites explains the important role of these genes coding for regulators of secondary metabolites and describes the mechanism at the transcriptomic level.

Plant crude extracts and partially purified plant extracts contain a mixture of active ingredients that differ in their chemical structure and composition, which in turn may affect their biological functions. The mechanism of action of plant extracts is related to their main components, which account for their defense properties against microbial agents as well as other biological properties. Although the exact mechanisms of antimicrobial and anti-toxigenic effects of plant-based bioactive compounds are not clearly defined; multiple, dose-dependent mechanisms have been reported. At high antimicrobial concentrations, plant bioactive compounds cause membrane damage, loss of energy production, enzyme dysfunction, and leakage of cell contents, leading to impaired cell physiology and cell death [14,41]. In addition, research has revealed that plant bioactive compounds at low and non-inhibitory concentrations affect pathogens by modulating gene transcription and protein expression [42,43,44,45,46,47,48]. Thus, the anti-toxicogenic activity mechanism of the plant extracts and complete inhibition of aflatoxin B1 production by the addition of *H. persicum*, *P. harmala* and *T. ammi* can be explained by their effective compounds and the gene transcription results.

Identification of herbal extracts capable of downmodulating the expression of genes but not affecting fungal physiology could be a promising strategy as it may avoid toxin synthesis without any impact on biodiversity and no risk of resistance [18]. Along this line, we found *H. persicum* (4 mg/mL), *P. harmala* (6 mg/mL) and *T. ammi* (2 mg/mL) to have a high potential to control aflatoxin B1 production, while they had no significant effect on growth of *A. flavus* and thus no significant impact on fungal biodiversity either [25]. As noted earlier, the inhibitory property of these extracts has been attributed to their phytochemicals. Phytochemical analysis of *H. persicum* has identified several groups of natural chemicals that can explain its inhibitory properties. These chemicals include volatile compounds (aliphatic esters, carbonyls, phenyl propyls and terpenes) and non-volatile compounds (flavonoids, furanocoumarins, tannins and alkaloids) as well as various minerals [49]. The main phytochemicals of *P. harmala* include alkaloids, flavonoids and anthraquinones. Among these compounds, alkaloids: mainly β-carbolines (harmin, harmalin, harmalol, harmol) and tetrahydroharmine were found to be the main substances responsible for the antimicrobial, antitumor and other biological activities [50]. The *T. ammi* seeds have been shown to possess various antifungal, antioxidant, and antimicrobial activities, which can be explained by the presence of various phytochemical constituents, mainly glycosides, saponins, phenolic compounds, and essential oils (e.g., thymol, γ-terpinene, cymene) [51,52,53].

In the present study, we selected three different concentrations of eight plant extracts according to their active compounds and antimicrobial effects that have been already shown in previous studies [51,54,55,56,57,58,59,60,61,62,63,64]. However, the effective concentration of herbal compounds may be different from the initial laboratory results, depending on the composition of the food and their physicochemical properties. For example, the amount of protein, fat and water activity in food can interfere with the antimicrobial activity of extracts, and hence there is often a need to use a different dose [65]. In addition to antimicrobial properties of the selected extracts, we considered their organoleptic characteristics to be used as a food or feed additive, as well as their affordability. In this regard, there are studies on the effect of herbal compounds, where a significant reduction in aflatoxin contamination due to the use of herbal extracts as a coating on nuts has been found. In a practical study by Kalli et al., it was shown that consumers were less exposed to aflatoxin B1 when *Cistus incanus* extract was added to macadamia. Their study showed that the *C. incanus* extract inhibited aflatoxin B1 by *A. parasiticus* in YES medium (87.1–90.1%) and in macadamia (72.5–85.9%) [66]. Another study showed that separate spraying of three herbal plant extracts including cinnamon, clove and celak on pistachio kernels could completely inhibit the growth of toxigenic fungi and biodegrade the produced aflatoxin when the nuts should be stored for more than one month [67].

Overall, the results of our study showed that the anti-aflatoxigenic mechanisms of the selected plant extracts occurred at the gene expression level. Particularly, we found that the effective doses of the selected plant extracts can control or block aflatoxin B1 production. These results highlight the potential of plant-based bioactive compounds in plant and crop protection.

## 4. Materials and Methods

### 4.1. Fungal Strain and Culture Conditions

The *Aspergillus flavus* strain NRRL 3357 was acquired from the collection of the Laboratory of Plant Pathology of Sapienza University. The strain was cultured on PDA at 28 °C for 7 days. Spore suspension was prepared in Tween 80 (0.05% in water). Spore density was calculated by hemocytometer to achieve a final concentration of 1 × 10^3^ conidia/mL.

### 4.2. Preparation of the Herbal Extracts and Determination of the Minimum Inhibitory Concentrations

The following medicinal herbs were selected for this study: Heracleum persicum, Peganum harmala, Crocus sativus, Trachyspermum ammi, Rosmarinus officinalis, Anethum graveolens, Berberis vulgaris, Berberis thunbergii. The plants used in this study were collected from their natural habitats in Iran, which were identified and confirmed in the Herbarium of Tehran University, and the herbarium code was issued for them (Table 1).

Dried plants were ground, then for each 100 g of the dried plants, 1000 mL of solvent (distilled water and ethanol) was added. For optimal extraction, these solvent ratios were selected based on polar and nonpolar ingredients of each herb (Table 1).

The herbs were macerated in the solvent and shaken at 220 rpm for 48 h. After filtering (Buckner funnel (90 mm diameter) with filter paper Whatman No.1), the extracts were concentrated using an evaporator and then dried by a freeze dryer.

Initially, we tested five different concentrations of each extract to determine the inhibitory concentration by measuring the dry weight of the fungal mass, and thereby three concentrations were selected for each extract. To prepare different dilutions of each extract, water and ethanol were used in accordance with the extraction ratios, where we used the maximum concentration of ethanol as a control to assure that it has no impact on fungal growth and aflatoxin production. For each concentration, three replications were considered and culture media after incubation were incubated for 7 days at 28 °C. The Yeast Extract Sucrose (YES) medium was used for testing aflatoxin production in this study.

### 4.3. Culture Media

To determine the best culture medium for aflatoxin production, six culture media were examined (Table 2). All media were prepared and sterilized by autoclaving at 121 °C for 20 min.

### 4.4. Quantification of A. flavus Hyphal Growth

Due to the differences in effective compounds of the plant extracts, the minimum inhibitory concentration (MIC) is different across them. To select the appropriate concentrations of the extracts for the main test, several pre-tests using different concentrations of each extract were performed on *A. flavus* and the growth rate of the fungal mass was measured. Given that we were primarily interested in inhibiting aflatoxin synthesis without a significant reduction in fungal growth, three concentrations of each extract that did not show a significant effect on fungal growth were selected (*p* > 0.05). To investigate the percentage of the fungal growth, 2 mL of sterile Yeast Extract Sucrose medium (YES) was placed in 15 mL glass tubes, and 3 different concentrations of the extract were added to each (Table 3). Next, glass tubes were shaken, inoculated with 40 μL of spore suspension, and then incubated in the dark for 7 days at 28 °C. After the incubation period, the mycelia were separated by Whatman filter paper grade No.1, were freeze-dried using an Alpha 1–4 LD plus freeze dryer, and subsequently their dry weights were measured. The percentage of hyphal growth was calculated by the following formula: $\frac{B}{A}\times 100$, where A denotes the dry weight of the control and B stands for the dry weight of the treatment with the tested extract.

### 4.5. Aflatoxin B1 Detection Using Liquid Chromatography–Mass Spectrometry (HPLC-MS/MS)

Aflatoxin B1 was extracted from the YES culture medium following the method reported in Ingallina et al., 2020, with some modification [68]. Two milliliters of the culture medium was mixed with equal volume of ethyl acetate and vortexed for 3 min twice and allowed to stand for 30 min, then centrifuged for 4 min at 10,000× *g*. The upper phase was transferred to a new vial. The ethyl acetate extracts were evaporated to dryness at 40 °C. The dried extracts were reconstituted in 100 μL methanol. The mixture was vortexed, centrifuged and then 80 µL of each sample was transferred to HPLC vials.

LC-MS analyses were carried out on an Agilent 1200 Infinity HPLC system coupled to an Agilent G6420 Triple Quadrupole mass spectrometer equipped with an Agilent API-Electrospray ionization source (Agilent Technologies, Inc., Santa Clara, CA, USA). The acquisition was in MRM positive ion mode [M+H]^+^. Chromatographic separation was performed with a Zorbax Eclipse XDB-C18, 50 × 4.6 mm inner diameter, 1.8 µm particle size, (Agilent Technologies, Inc., Santa Clara, CA, USA) at 25 °C and the injection volume was 5 µL. The mobile phases consisted of A: methanol/water/acetic acid 10:89:1 (*v/v/v*) and B: methanol/water/acetic acid 97:2:1 (*v/v/v*) while both contained 5 mM ammonium acetate. The total runtime was 20 min, with a flow rate of 0.4 mL/min. The gradient elution was as follows: 0–2 min 1%B, 3–14 min 99%B, 15–18 min 99% B, 19–20 min 1% B. The gradient was followed by 4 min for re-equilibration. Positive ESI-MS/MS was performed in MRM mode, AFLAB1 parameter analysis are: precursor ion 313.1, product ion 241.1, fragmentor 135 eV and collision energy 38 eV.

### 4.6. RNA Extraction and cDNA Synthesis

For the analysis of gene expression, the fungal mycelium was first washed in a cold Phosphate-Buffered Saline (PBS) buffer and then dried in a freezer-dryer. The freeze-dried mycelium was used. Extraction of RNA from the fungal mycelium was performed using TRIzol^®^ reagent (Invitrogen, Carlsbad, CA, USA) and according to the manufacturer’s instructions. Samples were treated with DNase I. To quantify RNA, a NanoDrop OneC Spectrophotometer (Thermo Fisher Scientific, Waltham, MA, USA) at a wavelength of 260 and 280 nm was used. To determine the quality of the extracted RNA, electrophoresis was performed using agarose gel (1.2% agarose). Total purified RNA was stored at −80 °C until further processing. Via reverse transcription, the cDNA was synthesized from 0.6–2 μg of RNA and random primers by using a RT-PCR kit (SensiFAST™ cDNA Synthesis Kit, Bioline Reagents, Ltd., London, UK).

### 4.7. Gene Expression Analysis by RT-qPCR

Primers were designed by The Primer3 software (version 0.4.0), validated using the Oligo analyzer 3.1 software, and synthesized by Sigma-Aldrich. The details of the primers are listed in Table 4. In order to determine the transcription of aflatoxin biosynthesis pathway genes, the Quantitative Real Time PCR reactions were carried out in a LineGene 9620 Bioer’s Real-Time PCR using a kit SYBR^®^ Premix Ex Taq™ (Takara Bio, Inc., Otsu, Japan) where β-tubulin was used as a reference gene and SYBR Green was used as a amplification detector. To specify the best concentration of primers, various concentrations in the range of 200–800 nm were tested. Amplification reactions were performed at a volume of 10 μL containing different concentrations of cDNA according to each condition, 5 μL of SYBR Green Supermix, 0.5 μL of each primer, and Milli Q Water up to 10 μL. Three replicates of each reaction were performed. PCR conditions for each gene are listed in Appendix A. In this study, fold change levels of the genes were evaluated using the 2^−ΔΔCT^ method. The reaction conditions and temperature pattern were as Table 4. All experiments were performed in triplicate and for gene expression assays, six biological replicates were used each time for each gene.

### 4.8. Statistical Analysis

#### 4.8.1. *A. flavus* Mycelial Growth and Aflatoxin B1 Production in Various Culture Mediums

To examine whether different culture mediums significantly affect the aflatoxin B1 production and the fungal growth, we used multivariate analysis of variance (MANOVA), where we specified aflatoxin B1 production and the fungal growth as dependent variables and the culture medium as an independent variable. The culture medium was defined as a categorical variable. Next, to find the culture medium that offers an optimal environment for the fungal growth and aflatoxin B1 production, we carried out analysis of variance (ANOVA) on each of fungal growth and aflatoxin B1 production, separately, as a function of culture medium. The culture medium was specified as a categorical variable.

#### 4.8.2. Effect of the Extracts on *A. flavus* Mycelial Growth and Aflatoxin B1 Production

On the selected culture medium, in which no extract was added, we measured the level of *A. flavus* mycelial growth and aflatoxin B1 production and used it as the control medium for each test. For each of the treatment conditions (different extracts and doses), we performed a two-sample *t*-test between the various concentrations of that extract and the control medium, to quantify the impact of concentration level on *A. flavus* mycelial growth and aflatoxin B1 production. The *t*-test was applied across repeated measurements of each extract and dose.

Subsequently, we applied a post-hoc analysis to examine whether the inhibition of aflatoxin production as a response to the addition of different doses of plant extracts is independent of the effect of fungal growth. For this purpose, we used a general linear model (GLM), where we specified the aflatoxin B1 production as a dependent variable, and a categorical variable (comprising two levels: “control”, and “treatment”) plus fungal growth as independent variables (aflatoxin B1 production ~ groups + fungal growth). This analysis was applied between each treatment condition and the control.

#### 4.8.3. The Impact of Plant Extracts on the Expression Level of Genes Involved in Biosynthesis Pathway of Aflatoxin B1

Like the previous analysis, we used the culture medium with no extract as a control, and measured the expression of three genes: *aflM*, *aflP*, and *aflR*. To quantify the impact of the addition of the different extracts with different concentrations on expression level of each gene, we applied a two-sample *t*-test between each treatment condition and the control medium.

#### 4.8.4. Correlation Analysis between the Gene Expression and Aflatoxin Production

To quantify the relationship between the aflatoxin production and the production of *flM*, *aflP*, and *aflR* genes, we applied Pearson correlation.

For all statistical analysis we checked the confidence level for three levels (* *p* < 0.05, ** *p* < 0.01, *** *p* < 0.001). We used the R software for statistical analyses and the “ggplot2” package for visualization.

## 5. Conclusions

Our study shows that biological approaches based on the addition of plant compounds can be relatively safe and effective natural agents against aflatoxin B1 contamination during the pre- and post-harvest, and demonstrates that effective doses of the selected extracts could be used to inhibit the aflatoxin. At the transcriptomic level, our study investigated two structural genes (*aflM* and *aflP*) and a regulatory gene (*aflR*) involved in the aflatoxin B1 biosynthesis, and found stronger and more consistent engagement of the structural genes. Future studies may explore engagement of other genes.

Furthermore, since we selected the eight medicinal plants on the basis of their organoleptic properties, and their affordability (in addition to anti-aflatoxigenic features), they could have applications in the food industry, particularly as an additive to human and livestock food that is prone to infection by toxin-producing fungi. Besides that, due to the increasing demand for natural foods, identifying and introducing a broad spectrum of effective natural compounds could be useful to reduce the consumption of synthetic preservatives.

## Figures and Tables

**Figure 1 antibiotics-09-00728-f001:**
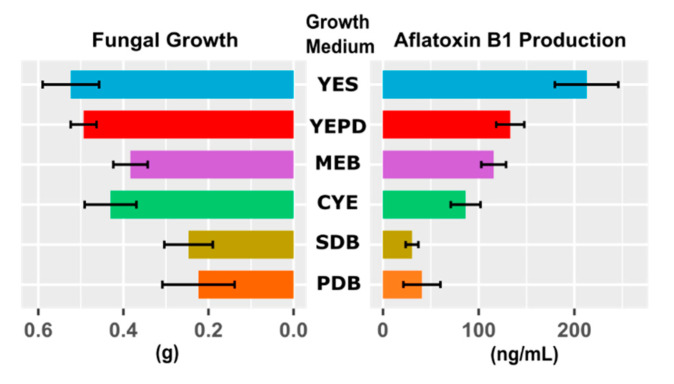
Selecting the optimal culture medium for *A. flavus* growth and aflatoxin B1 production. Mean and standard deviation (computed across repeated measurements) of *A. flavus* strain NRRL 3357 growth and aflatoxin B1 production in six culture mediums: Potato Dextrose Broth (PDB), Sabouraud Dextrose Broth (SDB), Czapek Yeast Extract (CYE), Malt Extract Broth (MEB), Yeast Extract Peptone Dextrose (YEPD), and Yeast Extract Sucrose (YES). Our analysis showed that, compared to other culture media, the Yeast Extract Sucrose (YES) medium provides a highly significant condition for fungal growth and aflatoxin B1 production.

**Figure 2 antibiotics-09-00728-f002:**
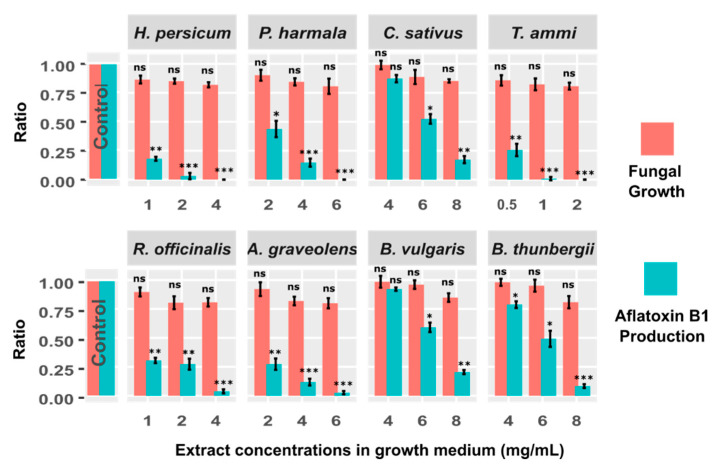
The effect of different concentrations of plant extracts (8 extracts in total) on growth of *A. flavus* NRRL 3357 and production of aflatoxin B1. Each bar shows the percentage of fungal growth (red bars) and aflatoxin B1 production (green bars) compared to control. To statistically assess the impact of these extracts on aflatoxin B1 production, we compared the aflatoxin B1 production between each of the treatment conditions (different plant extracts and doses) and the control using a two-sample *t*-test. Corresponding *p*-values are depicted above each bar (* *p* < 0.05; ** *p* < 0.01; *** *p* < 0.001; ns: not significant).

**Figure 3 antibiotics-09-00728-f003:**
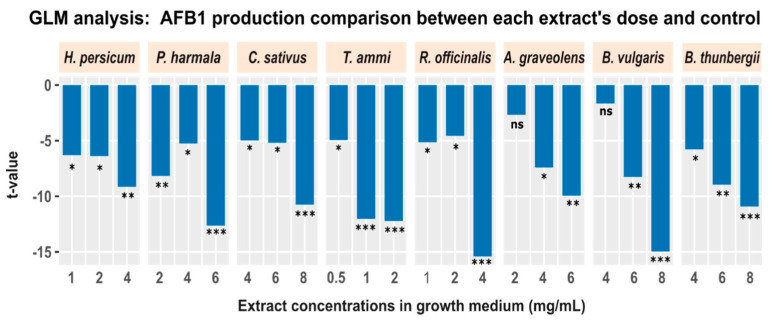
The effect of different concentrations of plant extracts on production of aflatoxin B1 after controlling for the potential impact of fungal growth. Each bar shows the comparison of aflatoxin B1 between all treatment conditions (different extracts and doses) and control, using a GLM to regress out the effect of fungal growth (* *p* < 0.05; ** *p* < 0.01; *** *p* < 0.001; ns: not significant).

**Figure 4 antibiotics-09-00728-f004:**
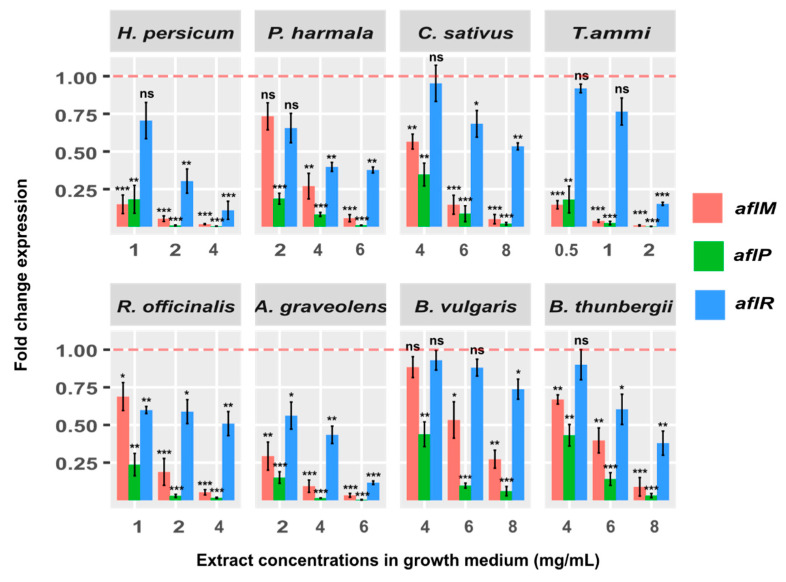
Fold change expression of *aflR*, *aflM* and *aflP* genes in *A. flavus* NRRL 3357 and in the YES growth medium after 7 days at 28 °C. The red line represents the control expression level. Each bar shows the fold change expression of the genes for all treatment conditions relative to that of the control treatment, where control was contaminated with *A. flavus* and untreated with extract. For statistical assessment, we compared between the control test and all treatment conditions (different plant extracts and doses) using a two-sample *t*-test (* *p* < 0.05; ** *p* < 0.01; *** *p* < 0.001; ns: not significant).

**Figure 5 antibiotics-09-00728-f005:**
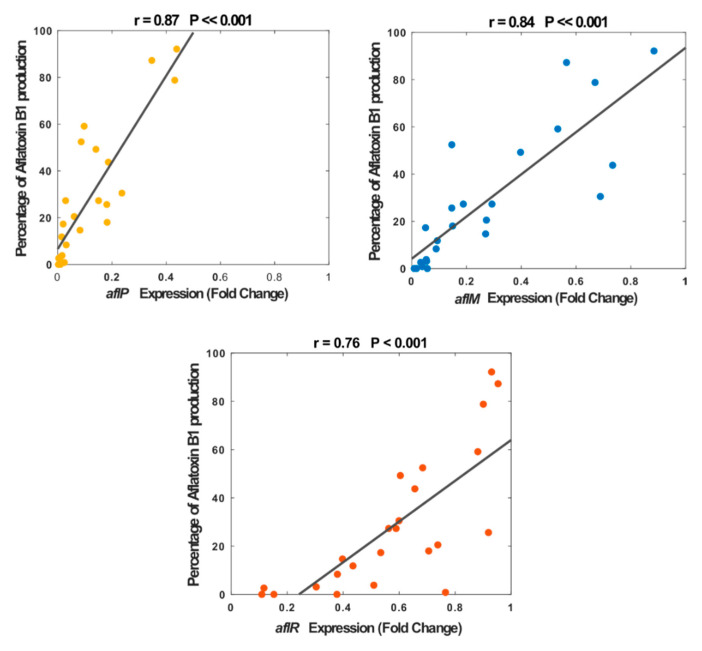
Relationship between the aflatoxin B1 production and the fold change of gene expression, for *aflM* (colored in blue), *aflP* (colored in yellow), and *aflR* (colored in orange) genes. Our results showed a significant correlation between the percentage of aflatoxin B1 production and the fold change expression of *aflM* (*r* = 0.84, *p* << 0.001), *aflP* (*r* = 0.87, *p* << 0.001), and *aflR* (*r* = 0.76, *p* < 0.001).

**Table 1 antibiotics-09-00728-t001:** Medicinal herbs and the solvent ratios used for the extraction process and herbarium code.

Common Name	Scientific Name	Solvent and Solvent Ratio	Herbarium Code
Persian hogweed	*Heracleum persicum*	Water/Ethanol (50/50)	PMP-762
Harmel	*Peganum harmala*	Water/Ethanol (70/30)	PMP-763
Saffron	*Crocus sativus*	Water/Ethanol (70/30)	PMP-524
Ajwain	*Trachyspermum ammi*	Water/Ethanol (50/50)	PMP-764
Rosemary	*Rosmarinus officinalis*	Water/Ethanol (50/50)	PMP-418
Dill	*Anethum graveolens*	Water/Ethanol (50/50)	PMP-356
Barberry	*Berberis vulgaris*	Water/Ethanol (70/30)	6879(THE)
Red Barberry	*Berberis thunbergii*	Water/Ethanol (70/30)	6880(THE)

**Table 2 antibiotics-09-00728-t002:** The culture media and their compounds.

Culture Medium	Composition
Potato Dextrose Broth (PDB)	(20 g Dextrose and 4 g Potato starch) for 1 L of media
Sabouraud Dextrose Broth (SDB)	(40 g Dextrose (Glucose) and 10 g Peptone) for 1 L of media
Czapek Yeast Extract (CYE)	(30 g Sucrose, 5 g Yeast extract, Czapek concentrate 10 mL and 1 g K2HPO4) for 1 L of media
Malt Extract Broth (MEB)	(17 g Malt Extract, 3 g Peptone) for 1 L of media
Yeast Extract Peptone Dextrose (YEPD)	(10 g Yeast Extract, 20 g Dextrose, 20 g Peptone) for 1 L of media
Yeast Extract Sucrose (YES)	(20 g Yeast Extract, 20 g Sucrose, 1 g Potassium dihydrogen Phosphate and 0.5 g Magnesium sulphate) for 1 L of media

**Table 3 antibiotics-09-00728-t003:** Selective concentrations for each extract.

Common Name	Plant Parts	Scientific Name	Concentration (mg/mL)
**Persian hogweed**	Seed	*Heracleum persicum*	1, 2, 4
**Harmel**	Seed	*Peganum harmala*	1, 2, 4
**Saffron**	Stigma	*Crocus sativus*	4, 6, 8
**Ajwain**	Seed	*Trachyspermum ammi*	0.5, 1, 2
**Rosemary**	Leaves	*Rosmarinus officinalis*	1, 2, 4
**Dill**	Leaves	*Anethum graveolens*	2, 4, 6
**Barberry**	Branches and Leaves	*Berberis vulgaris*	4, 6, 8
**Red Barberry**	Branches and Leaves	*Berberis thunbergii*	4, 6, 8

**Table 4 antibiotics-09-00728-t004:** The genes in the aflatoxin biosynthetic pathway and the primers.

Gene	Primer Pairs	Primer Sequence (5′-3′)	Annealing Temperature (°C)	PCR Product Size (bp)
*aflR*	F	CACCCCCTTGCGATTAGTGT	60.04	167
	R	GTTGATCGATCGGCCAGTCT	59.90	
*aflM*	F	TGGTGAACTACGCCCATTCC	60.04	137
	R	CACCGTCTCCGCCATTAACT	60.11	
*aflP*	F	CAGAGCGTCCGAATCCCTTT	57.30	141
	R	GGTAGACCTCTCCTTCCCGT	57.17	
*tubulin beta*	F	GTGACCACCTGTCTCCGTTT	59.89	211
	R	GGAAGTCAGAAGCAGCCATC	58.62	

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
