# Peer review of "The Potential of Plant-Based Bioactive Compounds on Inhibition of Aflatoxin B1 Biosynthesis and Down-regulation of aflR, aflM and aflP Genes"

_antibiotics, 2020, doi:10.3390/antibiotics9110728_

Round 1

Reviewer 1 Report

  1. Authors explained the advantage and output of these plant extracts for aflatoxin biosynthesis because already authors stated so many reports for aflatoxin biosynthesis such as El Khoury et al. (2017), Caceres et al. (2017), etc.
  2. Authors should explain the mechanism of action of three plant extracts H. persicum, P. harmala and T. ammi
  3. Authors should explain fold change levels affected regulatory factors along with control expression with figure
  4. Authors should discuss more on RT PCR expression profile analysis of genes regulating Aflatoxin B1 biosynthesis in A. flavus
  5. Authors should discuss transcriptomic effect of H. persicum, P. harmala and T. ammi on the expression of genes coding for regulators of secondary metabolites
  6. Authors should explain fungal enzymatic activities
  7. Authors should include microscopic images of A. flavus
  8. Line 268: in vitro and in vivo should be in italics.

Table 5 For PCR condition should be included in suppl. materials.

Regarding LC-MS analysis, few lines must be included in the discussion section. There must be included few lines

Reference 37: The scientific name should always be in italics.

Reviewer 2 Report

Line 20: The "Aspergillus flavus" should be italic.

Line 90: stop the aflatoxin B1 production.

As we all known, YES medium is good for fungal growth and aflatoxin B1 production compared with other media. So, I think that this part of study is unnecessary.  

The concentrations of different plant extracts are different. How to select the concentrations? Please provided more detailed information.

I think the followed conclusion is not correct, "This result may suggest a more relevant and robust role of structural genes in the mechanism of aflatoxin B1 inhibition by extracts.". Because aflR is transcriptional factors and very important for aflatoxin B1 production. Compared with structural genes, aflR expression is more stable.

Round 2

Reviewer 1 Report

According to me manuscript can be accepted in the current form

Reviewer 2 Report

It is accepted in the present form.